# CONVERT AND SPEAK: ACCENT CONVERSION WITH MINIMUM SUPERVISION

## ABSTRACT

Accent conversion(AC) aims to convert speech from one accent to another by changing the pronunciation pattern and prosody of source speakers while preserving linguistic content and speaker identity. This problem is quite challenging since 1) the parallel data with the same speaker speaking the same content in different accents rarely exists; 2) Accents can result in differences not only in prosody but also in the pronunciation units for some accents. In this work, we propose a conversion-and-speaking framework based on speech generative models to tackle this problem. The accent conversion is achieved by converting the source semantic tokens to the target ones. Specifically, a separate sequence-to-sequence task based on an autoregressive decoder-only transformer has been designed to accomplish the conversion. Conditioned on the converted semantic tokens, a speech generative model based on TF-Codec, a neural speech codec trained with large amounts of target accent speech, has been proposed to generate speech with target accent prosody. Unlike multi-stage generation used in other generative models, we use single-stage autoregressive generation thanks to the TF-Codec that uses a single-stage group quantization, which helps to reduce the complexity and latency of the generation process. To relieve the dependence on the parallel data, we pretrain the conversion module with a pretext task in a self-supervised manner using large amounts of target accent speech to learn the probability space of the target semantic tokens first so that small amounts of parallel data are needed to learn the mapping of specific accent pronunciation units with their targets. Experiments on Indian-English to general American-English conversion show that the proposed framework achieves state-of-the-art performance in accent similarity, speech quality, and speaker maintenance. With the pretraining, only 15 minutes of parallel data, which is not constrained to the same speaker, are required to achieve a good conversion quality. The proposed generative model also achieves higher speech quality and speaker similarity with lower complexity and latency (50 AR steps/1 sec of audio) than multi-stage speech generation methods (75 AR steps+7 NAR steps/1 sec of audio). With less supervision from parallel data, this framework can be easily extended to other accents with low-resource data. [1]

## 1 INTRODUCTION

Accent brings barrier of understanding when having a conversation between speakers in different accents. The technology of accent conversion aims to break such barrier to make the source accent speakers sound as target accent speakers by changing the pronunciation pattern and prosody while preserving the linguistic content and his/her own speaker identity. The challenges lie in that accent features are not only highly correlated with prosodic characteristics, e.g. speaking rate and duration but also pronunciation patterns, which result in different phonetic representations. In some early researches, they try to explicitly model such pronunciation patterns by building some accent-specific dictionaries to include all possible pronunciation of every word according to the accent type. These methods tend to have poor adaptation because of its assumption that phonetic knowledge about every accent is available and all speakers can be categorized into a few accent clusters(Huckvale, 2006).

---

[1]Audio samples are available at https://convert-and-speak.github.io/demo/

Conventional AC methods(Zhao et al., 2018a; 2019; Li et al., 2020; Ding et al., 2022) simplify this problem by just converting the speaker identity of a target accent speaker's voice to that of the source accent speaker's since the speaker identity is easier to model apart from prosody and content. These methods hinder their usage in real applications since the target accent reference is hardly available at conversion stage.

Therefore, reference-free AC methods are more feasible and promising. Some previous approaches (Zhao et al., 2021; Nguyen et al., 2022) try to learn the acoustic mapping between the source accent speech and target accent speech directly with the parallel data in which the same speaker speaks the same content with two different accents. However, such data is extremely rare. So the voice conversion(VC) technology is usually used to synthesize the data set by converting the speaker identity of the target accent speech to the source speaker's. This data-driven solution suffers from bad conversion quality with unnatural sound and bad speaker maintenance due to the lack of enough high-quality data and error introduced from VC stage.

To relieve the dependence of parallel data, another kind of approaches (Liu et al., 2020; Jia et al., 2022) which leverage disentanglement technology to remove accent from content, speaker identity, prosody and resynthesize to the target waveform through a syhthesizer, e.g. text-to-speech(TTS) model. The synthesizer is trained on the target accent speech to generate speech with prosody in target accent. To remove accent from content and speaker identity, some auxiliary models or tasks are carefully designed, e.g. accent-agnostic automatic speech recognition(ASR) model or phoneme classification task. Such non-parallel AC methods is not good enough in terms of prosody modeling and speaker maintenance. These methods are also hard to extend to any accent.

In this work, we regard accent conversion as a sequence-to-sequence task on the semantic token level and propose a new framework in which a separate conversion module is designed and a speech neural codec based generative model is used to synthesize the speech with target accent. Specifically, the semantic tokens are extracted by HuBERT(Hsu et al., 2021). To generate target accent speech with high-quality and low-cost, we design a single-stage autoregressive generative model based on TF-Codec(Jiang et al., 2023) and train it with abundant target accent speech. To relieve the burden on the parallel data, we use large amount of target accent speech which is much more easier to get to build the correlation of semantic tokens in the target accent domain with pretraining tasks and finetune the conversion module with a small amounts of parallel data.

The proposed framework is capable of both phoneme and prosody conversion. Extensive experimental results on public Indian-English to general American-English conversion dataset show the proposed framework achieves the state-of-the-art performance in terms of accent similarity, speech quality and speaker maintenance.

With less supervision from parallel data and simple design, this framework can be easily extended to other low-resource accents with limited data. The main contributions of this work include:

- A new framework composed of a conversion and speaking module with a speech nerual codec based generative model has been proposed for accent conversion tasks. The accent conversion can be achieved at the semantic token level. Extensive results show its superiority over existing AC methods and generative-only methods.

- For limited parallel data available in AC task, instead of acoustic-mapping based methods, we can leverage the pretraining technology to learn the probability space of the target accent domain with large amounts of target accent speech available and use small amounts of parallel data to learn the mapping of specific pronunciation units with their targets. Experimental results show that only 15 minutes of parallel data, not constrained to the same speaker, can achieve a good quality.

- We propose a new speech generative model based on TF-Codec, which generates codes in a single-stage pure-autoregressive manner. Experimental results show it achieves better speech quality and speaker similarity at lower computation cost and latency compared with the multi-stage generation process in other generative models based on Encodec (proposed: 50 AR steps/1 sec of audio vs Encodec-based: 75 AR steps+7 NAR steps/1 sec of audio).

## 2 BACKGROUND

For accent conversion task, there has not been a public parallel corpus that contains pairs of audios having the same contents yet coming from the same speakers in different accents. So mainly two kinds of methods are proposed to accomplish this task in the literature. One is to synthesize the dataset containing the pairs of audios in the same voice but in two different accents with another voice conversion model and learn the acoustic mapping between them to accomplish accent conversion.(Zhao et al., 2021) build the golden speaker utterance by converting the general American-English speaker's voice to the source-accent speaker's with a pretrained source speaker's synthesizer. Then use this golden speaker utterance as the target to learn the mapping of the mel-spectrogram based on a seq2seq VC system. (Nguyen et al., 2022) use a pretrained VC model to build the parallel data and trained the AC model based on Tacotron(Shen et al., 2018) conditioned on the semantic representation extracted from wav2vec 2.0(Baevski et al., 2020). This data-driven approch needs large amounts of data to achieve a good zero-shot ability and the auxilary VC model usually needs to be finetuned on the AC data set to alleviate the error caused by voice conversion step. These methods also constrain the output to be generated with the same length of the input which limits the conversion quality since the prosody of the speech is largely affected by the accent.

Another approach is to regard accent conversion as a decomposition and resynthesis task in which the accent is separated from content, speaker identity, prosody and resynthesize to the target waveform in a TTS manner. (Liu et al., 2020) disentangles different features in multi-stage with several off-the-shelf models. Specifically, an accent-robust ASR model is trained using source accent speech with text labels to separate the accent from the content. A multi-speaker TTS model with a global speaker encoder is trained with large corpus of target-accent speech to map the accent-agnostic linguistic features to acoustic features with the voice of source speaker and target accent prosody. Another work (Jia et al., 2022) treats the decomposition and resynthesis in an end-to-end manner. It designs a Pseudo Siamese Disentanglement Network (PSDN) with two streams in which one stream is used to learn the acoustic feature of target accent speech and the other auxiliary stream is used to build the information gap with the target stream to disentangle the content with accent, complemented with another adversarial accent classifier with gradient reversal layer(GRL). Such kind of approach relieves the burden on the parallel data but need to design the auxiliary tasks carefully. The performance of disentanglement is hard to evaluate and explain. What's more, the mapping-based synthesis models have been largely surpassed by generative-based synthesis models recently in terms of zero-shot ability and speech quality.

## 3 ACCENT SPEECH ANALYSIS

As previous study(Huckvale, 2006) shows, the accent feature affects speech in many aspects such as intonation, rhythm and pronunciation patterns. Take Indian-English for example, they may pronounce $'v'$ as $'w'$ or vice versa, $'th'$ as $'t'$ or $'d'$ and $'p'$ as $'b'$. Besides such pronunciation units difference, the prosody, e.g. intonation, stress is also changed a lot according to the accent. We compare the pitch contour of the general American-English and Indian-English when they read the same sentence in Figure 2 in Appendix A.2. It can be seen that the intonation, rhythm, e.g. duration of each word and stress are quite different, in which Indian-English tends to speak with more ups and downs and the general American-English sounds more monotonous.

In the proposed framework, the pronunciation patterns are converted at discrete semantic token level and the prosody is converted with a pretrained generative model. For semantic representation, we use HuBERT as a semantic feature extractor, which generates a discrete semantic token sequence at 50Hz framerate for 16kHz audio. Previous studies(Polyak et al., 2021; Huang et al., 2022) show that HuBERT is a good representation for speech content and removes most of the speaker identity, verified by good performance in voice conversion task. But there is no clear clarification on how HuBERT reacts on the accent speech. In Section 3.1, we are going to show our observations on the effect of the accent on HuBERT tokens.

For prosody conversion, we use a speech neural codec based generative model, in which a style prompt is used to control the speaker identity through the in-context learning. To maintain the source speaker's identity, we use the source accent speech as the prompt during inference. This need

to be verified if accent feature will be caught through the in-context learning, which is discussed in Section 3.2.

## 3.1 HuBERT TOKENS ON ACCENT SPEECH

In this section, we evaluate how accent affects the HuBERT tokens. We employ the HuBERT-Base(Hsu et al., 2021) model[2] which is trained on the general American-English data and k-means algorithm with 500 clusters to extract semantic tokens. We take general American-English as target accent and Indian-English as source accent. Specifically, we build a parallel data set from L1-L2 ARCTIC dataset (Kominek & Black, 2004; Zhao et al., 2018b) in which the Indian-English speaker and general American-English speaker speak the same content. Both of them are fed into HuBERT to get the semantic token sequence. Then we use the metric Longest Common Subsequence(LCS) to evaluate the similarity of the two sequences. To eliminate the disturbance of the duration of each word, the duplicated tokens in the sequence are removed. Since the length of each audio is different, the Longest Common Subsequence Ratio ($LCSR = LCS/utterance\_length$) is used. The audio length of the parallel data in each pair is similar, ranging from 3 seconds to 5 seconds. The smaller utterance length is used to calculate the LCSR of each pair. 1000 pairs are used in this experiment. To further study the phoneme change effect, the cases are divided into the one with phoneme change and without phoneme change. Since the speakers of each pair are different, the effects of LCSR on the speaker identity is also calculated as a reference.

The results are shown in Table 1 with $5\%$ confidence interval. As the results shown, with the source accent introduced, HuBERT tokens have changed a lot, degrade from 0.747 to 0.569 in terms of LCSR. For those cases with specific phoneme changes, more tokens have been changed from the target accent references(LCSR: 0.541).

Table 1: HuBERT tokens on accent speech. Lower LCSR means less similarity between source accent speech and target accent speech. Source accent: Indian-English. Target accent: general American-English.

| Influencing factors | LCSR(p=0.05) |
|---|---|
| Speaker identity | 0.747±0.003 |
| Accent without phoneme change(w. speaker identity change) | 0.569±0.003 |
| Accent with phoneme change(w. speaker identity change) | 0.541±0.003 |

## 3.2 IN-CONTEXT LEARNING WITH ACCENT PROMPT

In this section, we evaluate if the accent feature will be extended through the in-context learning. The idea is to design an A/B testing to compare the accent similarity of the synthesized speech generated with two kinds of prompt in different accents conditioned on the same content. For testing, we take Indian-English and general American-English as prompt type. The generative model proposed in this paper which is trained on general American-English speech is used to generate speech according to the testing prompt. Specifically, we build 100 pairs of samples to test. Each pair contains an utterance in general American-English which is used to extract HuBERT semantic tokens, an utterance from a general American-English speaker and from an Indian-English speaker. For subjective testing, 20 participants who are college students majored in foreign language are asked to distinguish the two synthesized speech and choose the one sounds more close to general American-English. The probability of being selected is used as the evaluation metric. The results are shown in Table 2. The probability of each prompt type is about 0.5, which indicates the synthesized speech with different accent prompts are hard to distinguish and both close to general American-English. This means the accent style can not be transferred through the in-context learning. So the source accent speech can be used as an prompt to catch the source speaker's identity without bringing the source accent back to the converted speech.

---

[2]https://huggingface.co/facebook/hubert-base-ls960

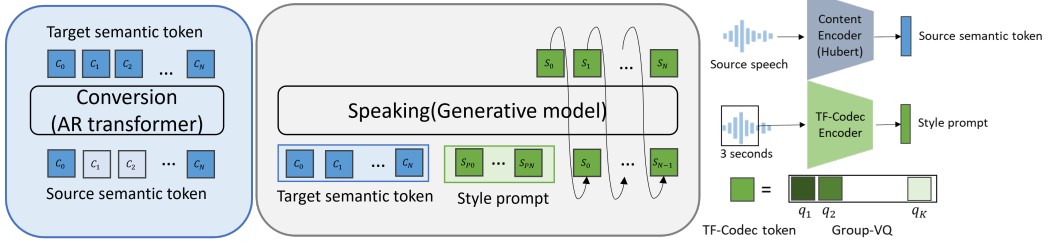

Figure 1: Proposed framework. The source accent semantic tokens are converted to target accent semantic tokens first and generate speech with target accent prosody conditioned on the converted semantic tokens and the style prompt from the first 3 seconds of the source speech. TF-Codec token is a group of concatenated embeddings of each quantizer.

Table 2: Comparison of the synthesized speech with general American-English and Indian-English prompt

| Prompt type | Probability of being selected as general American-English |
|---|---|
| General American-English | 0.5 |
| Indian-English | 0.5 |

## 4 PROPOSED FRAMEWORK

### 4.1 OVERVIEW

According to the preliminary analysis on the accent speech, the accent feature has a large influence on the HuBERT tokens. So we introduce a separate conversion module to convert the source-accent HuBERT tokens to the target-accent ones. Also, we find the accent style cannot be caught through the in-context learning so that we can use the source-accent speech as style prompt directly in the generative model to maintain the speaker identity. The overall structure is shown in Figure 1. Both the conversion and generative model are based on autoregressive decoder-only transformer structure. More details are discussed in Section 4.2 and Section 4.3.

### 4.2 SOURCE-ACCENT SEMANTIC TOKEN CONVERSION

The conversion module is designed as a sequence-to-sequence task in HuBERT token space in which the source-accent semantic tokens are converted to the target-accent semantic tokens iteratively in an autoregressive manner. To handle the shortage of parallel data, we use large amounts of target-accent English data to pretrain the conversion module with a pretext task as BART and T5(Lewis et al., 2019; Raffel et al., 2020). We then finetune the conversion module with a small amount of parallel data. Note that the parallel data is the same content spoken by different speakers.

**Pretraining** The pretext task aims to build the probability space of HuBERT tokens in the target-accent domain so that the target-accent semantic tokens can be generated according to its context of previous tokens in the target-accent domain. In this pretext task, the model is trained in a self-supervised manner which is to produce the original token sequence $Y = \{y_0, ...y_t\}, t < T$ conditioned on the corrupted token sequence $\bar{Y} = \{\bar{y_0}, ...\bar{y_t}\}, t < T$, formulated as

$$p(Y|\bar{Y}; \theta_{AR}) = \prod_{t=0}^{T} p(y_t|y_{<t}, \bar{Y}; \theta_{AR})$$  (1)

We have experimented with corruptions like token masking, token deletion, and token infilling and we find the masking scheme works the best. Specifically, following the masking scheme in

BERT(Devlin et al., 2019), 15% tokens have been randomly selected first. Then 80% of them are replaced with [MASK] tokens, 10% of them are filled with random tokens and the remaining 10% are left unchanged. We train the pretext task with large amounts of target-accent data which is available in the public corpus.

**Finetuning** Since some phonemes in the source-accent need to be converted to the target-accent ones, a mapping between these phonemes need to be learned. Specifically, we finetune the pretrained conversion model with a small amount of parallel accent data. Besides the phoneme conversion, we find such stage can also help to convert the prosody more closely to target-accent. Correspondingly, the training can be formulated as

$$p(Y|X; \theta_{AR}) = \prod_{t=0}^{T} p(y_t|y_{<t}, X; \theta_{AR}) \tag{2}$$

in which $X = \{x_0, ...x_t\}, t < T$ is the source-accent semantic token sequence and $Y = \{y_0, ...y_t\}, t < T$ is the target-accent semantic token sequence.

### 4.3 TARGET-ACCENT SPEECH GENERATION

The target-accent speech generation is achieved by training a separate generative model on a large target-accent speech corpus. This model generates acoustic tokens of TF-Codec(Jiang et al., 2023) iteratively through a single-stage causal speech generation, conditioned on the (converted) target-accent semantic tokens.

#### 4.3.1 SPEECH TOKENIZER WITH TF-CODEC

We use the pretrained causal neural speech codec TF-Codec to extract the acoustic token of each frame. Unlike Jiang et al. (2023), we remove the predictive loop and use the non-predictive model at 6 kbps for efficient acoustic modeling with high quality output. Specifically, the TF-Codec takes the 16kHz magnitude-compressed time-frequency spectrum with a window length of 20 ms and a hop length of 5 ms as input. Then a stack of 2D causal convolutional layers, followed by a temporal convolutional module (TCM) and a gated recurrent unit (GRU) block is used to capture the short-term and long-term temporal dependencies between the input frames in a causal manner. For the quantization, it combines 4 frames together, producing a frame rate of 50 Hz for quantization. Instead of using RVQ, it employs group quantization where the latent embedding is splitted into $K$ groups and each group is quantized by a vector quantizer with a codebook of 1024 codewords. All $K$ acoustic codes are concatenated and decoded to get the reconstructed waveform.

#### 4.3.2 SINGLE-STAGE CAUSAL SPEECH GENERATION

As the group quantization in TF-Codec encodes each group independently, we leverage a single-stage causal speech generation to generate acoustic codes of all $K$ quantizers simultaneously for each frame. As shown in Figure 1, TF-Codec token, which is the concatenated embeddings corresponding to all $K$ quantizers, is generated in one-stage autoregressive manner conditioned on the target-accent semantic tokens and style acoustic tokens. For each group embedding in TF-Codec token, the dimension is $D_{token}/K$, in which $D_{token}$ is the dimension of the embedding in transformer. $K$ classification heads are employed to predict the $K$ acoustic codes for current frame separately. The training target can be formulated as

$$p(C_:|Y, \widetilde{C}_:; \theta_{AR}) = \prod_{t=0}^{T} p(c_t|c_{<t}, Y, \widetilde{C}_:; \theta_{AR}) \tag{3}$$

in which $Y = \{y_0, ...y_t\}, t < T$ is the semantic token sequence from target-accent speech. $C_:$ is TF-Codec token sequence of target-accent speech. $\widetilde{C}_:$ is the TF-Codec token sequence of style acoustic prompt. We do not distinguish $\widetilde{C}_:$ from $C_:$ in training. The concatenation of $\widetilde{C}_:$ and $C_:$ is a whole sequence. During inference, the first 3 seconds of the source speech is used as $\widetilde{C}_:$.

## 5 EXPERIMENTS

To evaluate the performance of the proposed framework, we take Indian-English as source accent and general American-English as target accent, a common scenario in the research literature.

### 5.1 EXPERIMENTAL SETUP

**Dataset.** For the generative model and pretraining stage of the conversion model, LibriTTS dataset (Zen et al., 2019) is used as our training data. The dataset contains approximately 585 hours of general American-English speech data, sourced from audiobooks available on the public LibriVox project. For the finetuning stage of the conversion model, L1-L2 ARCTIC dataset (Kominek & Black, 2004; Zhao et al., 2018b) is used. L1-L2 ARCTIC dataset is a dataset with accent speakers speaking the same content. To build the parallel data, we select a general American-English speaker named "bdl" as the target-accent speaker and "ASI" as the Indian-English speaker. Among all their utterances, 1000 utterances, about 50 minutes of speech are used in the training, 50 utterances are used in validation and the remaining 100 utterances are used for testing. To better verify the zero-shot ability, we also add speaker p248 from VCTK dataset and another 4 Indian speakers from L1-L2 ARCTIC dataset into testing. To be noted that the 20 utterances from speaker p248 in VCTK are used to compare with the existing machine-learning based AC method (Liu et al., 2020). Besides these 20 cases, we add random 20 utterances of each testing speaker in L1-L2 ARCTIC for objective evaluation, e.g. 120 cases in total, and random 8 utterances per speaker for subjective evaluation, e.g. 60 cases in total.

**Model and configuration.** The number of quantizers in TF-Codec ($K$) is set to 16. The transformer used in conversion model and generative model are the same structure with 12 layers of 16 attention heads, a feed-forward layer with dimension of 4096, and a dropout layer with rate of 0.1. The embedding dimension in transformer($D_{token}$) is 1024. The generative model and pretraining stage of conversion model are trained on 8 NVIDIA TESLA V100 32GB GPUs with a batch size of 5k tokens per GPU. The ScaledAdam(Yao et al., 2023) optimizer is used. The learning rate is set to 0.01, with a warmup for the first 5k steps and decays exponentially. The generative model is trained for 500k steps and the conversion model is trained for 100k steps. The finetune stage of the conversion model is processed on one GPU of NVIDIA Tesla A100 80GB, with a batch size of 20k tokens. The same optimizer is used. The learning rate is set to $2 \times 10^{-5}$, with a warmup for the first 160 steps. The finetuning of the model is trained for 1k steps. During inference, we employ Top-$k$ algorithm to generate each token, in which $k = 2$ for conversion model and $k = 10$ for generative model. For each case, we inference for 5 times and select the one which performs the best.

**Baseline models.** To show the superiority of the proposed framework, we select 3 models as our baselines. The existing machine-learning based AC method (Liu et al., 2020), which is the best model available in the public to our knowledge. The generative-only models are also used as our baselines, in which we compare with the multi-stage generative model based on EnCodec and single-stage generative model based on TF-Codec.

**Evaluations.** To evaluate the speaker identity maintenance, the speaker similarity metric is calculated as the cosine similarity of the two speaker vectors, which are extracted from the source accent speech and the converted speech, correspondingly. WavLM-TDNN[3] (Chen et al., 2022), a state-of-the-art speaker verification model, is used to get the speaker vector from a speech. To evaluate the naturalness, We conduct MOS testing, in which the raters are asked to give a score ranging from 1 (lowest quality) to 5 (highest quality) according to the overall subjective quality. To evaluate the performance of accent conversion, the intuitive metric LCSR is used. We also conduct an A/B testing in which participants are asked to choose the one that sound more close to general American-English accent in each A/B pair. Particularly, the participants are trained to distinguish the accent difference by listening to a pair of <Indian-English, general American-English> sample at first. The probability of being selected as general American-English accent is used as the metric of the accent similarity. 20 participants who are college students majored in foreign language are invited to conduct these evaluations.

---

[3]https://github.com/microsoft/UniSpeech/tree/main/downstreams/speaker_verification

Table 3: Comparison on speaker p248 in VCTK dataset(20 cases)

| Framework | MOS-Naturalness(↑) | SPK(↑) | MOS-Accent(↑) |
|---|---|---|---|
| Ground truth(in different speakers) | 4.34 | - | 70.0% |
| Liu. et al (Liu et al., 2020) | 3.95 | 0.168 | 67.6% |
| Generative model(EnCodec) | 3.84 | 0.429 | 35.1% |
| Generative model(TF-Codec) | 4.00 | 0.502 | 35.0% |
| Proposed | 4.08 | 0.408 | 69.3% |

Table 4: Comparison on L1-L2 ARCTIC dataset. LCSR of ground truth speech is calculated between ground truth utterances of different speakers.

| Framework | MOS-Naturalness(↑) | SPK(↑) | MOS-Accent(↑) | LCSR(↑) |
|---|---|---|---|---|
| Ground truth(in different speakers) | 4.14 | - | 79.3% | 74.4% |
| Generative model(EnCodec) | 3.79 | 0.511 | 35.0% | 54.5% |
| Generative model(TF-Codec) | 3.91 | 0.543 | 35.2% | 54.5% |
| Proposed | 3.93 | 0.438 | 74.3% | 62.2% |

## 5.2 ACCENT SIMILARITY, SPEECH QUALITY AND SPEAKER SIMILARITY

**Accent similarity.** As shown in Table 3 and Table 4, the MOS-Accent metric on both datasets ranks the highest and very close to the ground truth. Compared with the generative-only models, e.g. Generative model(EnCodec) and Generative model(TF-Codec), the proposed framework highly surpasses them in terms of MOS-accent. This is also verified by the LCSR metric on L1-L2 ARCTIC dataset, where the proposed framework is closely approach the ground truth LCSR. The super-low MOS-Accent of generative-only models also indicates limited accent conversion ability without the conversion module. Compare the proposed with Liu. et al (Liu et al., 2020), Liu's method is capable of accent conversion but the speech quality and speaker similarity is much worse according to the MOS-Naturalness and SPK. Artifacts can be found in some cases, as shown in the demo page.

**Speech quality and Speaker similarity.** According to the MOS-Naturalness metric in Table 3 and Table 4, the proposed framework ranks the highest. What's more, we find the speech quality highly relies on the codec used in the generative model since the MOS-Naturalness of the proposed and Generative model(TF-Codec) are very close, both at the higher level, compared with the EnCodec-based generative model. The higher speaker similarity can also be achieved with the TF-Codec based generative model. This can also be verified by our demo cases. Compare the proposed with Generative model(TF-Codec), the SPK value drops a bit but the subjective judgement on the demo cases are quite similar. We think this is caused by the error from the speaker vector extractor WavLM-TDNN. Most probably, the speaker vector from WavLM-TDNN contains not only the speaker identity but also accent information. So with better accent conversion, the speaker vector of the converted speech and the accent source speech tends to be more different, results in a lower SPK. It should be more reasonable to use this metric to compare Generative model(EnCodec) with Generative model(TF-Codec) and the proposed with Liu's method since both of them are with/without accent leak in the converted speech.

## 5.3 EFFICIENCY OF SINGLE-STAGE CAUSAL SPEECH GENERATION

Here we compare the complexity of the proposed single-stage causal speech generation scheme based on TF-Codec with the multi-stage speech generation scheme based on Encodec. In the Encodec-based generative models, two stages are usually taken, with a combination of autoregressive(AR) stage to generate the first quantizer and non-autoregressive(NAR) stage to generate the rest of the quantizers of all time step based on the previous quantizers. The Encodec used in the experiment is composed of 8 quantizers with the frame rate of 75 Hz and sample rate of 24kHz. The complexity is shown in Table 5 in terms of model parameters and decoding steps. According to Table 5, TF-Codec based generative model saves more than 50% in model size and takes pure causal decoding scheme with less steps.

Table 5: Complexity comparison(Generative model(EnCodec) vs Generative model(TF-Codec))

| Framework | Model parameters(M) | Decoding steps(/s) |
|---|---|---|
| Generative model(EnCodec) | 262.3 | 75 AR + 7 NAR |
| Generative model(TF-Codec) | 100.8 | 50 AR |

## 5.4 TRAINING WITH MINIMUM SUPERVISION

In this section, we further reduce the parallel data used in the finetuning stage of the conversion model, from 50 minutes (proposed in Table 3 and Table 4) to 30 minutes and 15 minutes, respectively. We use the MOS-Accent as the evaluation metric and test on the VCTK test set. We also add the baseline models into A/B testing for comparison. As shown in Table 6, by decreasing the data amount, the performance drop is negligible. With the minimum supervision of 15 minutes, the performance is still superior to the baseline models, which shows its high potential for extension on the other accent with low-resource data, such as Chinese-English and Korean-English to general American-English cases in our demo page.

Table 6: Accent conversion quality with minimum supervision

| Framework | Parallel data amount | MOS-Accent(↑) |
|---|---|---|
| Liu. et al (Liu et al., 2020) | / | 56.2% |
| Generative model(EnCodec) | / | 19.3% |
| Generative model(TF-Codec) | / | 19.2% |
| Proposed | 50 mins | 59.3% |
| Proposed | 30 mins | 58.1% |
| Proposed | 15 mins | 56.8% |

## 5.5 ABLATION STUDY

In this section, we further study some important designs in the proposed framework to show the effectiveness of our choices. The results in terms of LCSR are shown in Table 7 on L1-L2 ARCTIC test set. Specifically, we compare with the solution in which the limited parallel data is used to finetune the pretrained generative model directly. This solution totally fails as the 2.0% LCSR in Table 7 shown, indicating hardly any accurate content preserved through this mapping-based learning method with little amount of parallel data. We also compare with another solution where the conversion model is trained from scratch with the small amount of parallel data. Without pretraining, the result degrades to a large margin.

Table 7: Ablation study on framework design

| Framework | LCSR(↑) |
|---|---|
| Proposed | 62.2% |
| Finetune on generative model | 2.0% |
| Conversion model without pretrain | 10.3% |

## 5.6 CONCLUSIONS

In this work, we regard accent conversion as a sequence-to-sequence task on the semantic token level and propose a new framework in which the source-accent semantic tokens are converted to the target-accent semantic tokens and synthesized to speech with target-accent prosody by a neural speech codec based generative model. Experimental results show the proposed framework achieves the state-of-the-art performance in terms of accent similarity, speech quality and speaker maintenance. With pretraining on target-accent speech, only 15 minutes of parallel data which is not constrained to the same speaker are required to achieve a good conversion quality. Furthermore, the proposed generative model based on TF-Codec achieves higher speech quality and speaker similarity with lower complexity and latency. With less supervision, this framework can be easily extended to other accents with low-resource data. We will further improve the evaluation framework of accent conversion task by enlarging the testing scale and a more normative subjective evaluation framework in our future research.

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

# A APPENDIX

## A.1 RELATED WORK ON SPEECH GENERATIVE MODELS

Recently, the generative models show large potential in generating contextual consistent, natural and diverse audio/speech with the in-context learning of a referenced prompt. AudioLM(Borsos et al., 2023), designed for zero-shot audio generation is the first work to show the strong ability of the in-context learning with a short prompt to maintain acoustic information such as speaker identity, prosody style and acoustic environment in the continuations. VALL-E(Wang et al., 2023a), verified in the TTS task has shown better zero-shot ability, speech naturalness and diversity than traditional TTS models.

With regard to the model structure, these models take the decoder-only auto-regressive transformer structure to build the correlation of the acoustic features tokenized by a neural speech codec conditioned on the semantic tokens. The neural codec codes are taken to the stage as an efficient intermediate representation of acoustic features for speech synthesis, containing rich information while in a compact space. AudioLM learns to predict the acoustic tokens extracted from Soundstream(Zeghidour et al., 2022) conditioned on HuBERT(Hsu et al., 2021)-like semantic tokens with multiple AR stages, in which the first several quantizer layers are predicted in the first stage to get the coarse information of the speech and the rest layers are predicted based on the coarse layers to get the fine details of the speech. This framework can be easily extended to other tasks such as voice conversion(Wang et al., 2023b) with the semantic token extracted from HuBERT and synthesized with the prompt of another speaker's utterance. VALL-E is a text conditioned generative model trained on Encodec(Défossez et al., 2022) tokens. It takes the hierarchical structure in which the first quantizer is generated with the AR model and the others are generated with the NAR model(all frames are predicted simultaneously when predicting each codebook) based on the previous quantizers. These models all take the multiple stages to generate the final speech since the neural codecs they used are built on residual vector quantization(RVQ) which consists of a hierarchy of all the vector quantizers. Encodec encodes each frame with 8 codebooks at a 75Hz frame rate. To generate 1 second of audio, it needs 75 (AR) steps with 1 code generated for each step and 7 (NAR) steps to generate 75 codes of all the frames for each step.

## A.2 ACCENT ANALYSIS

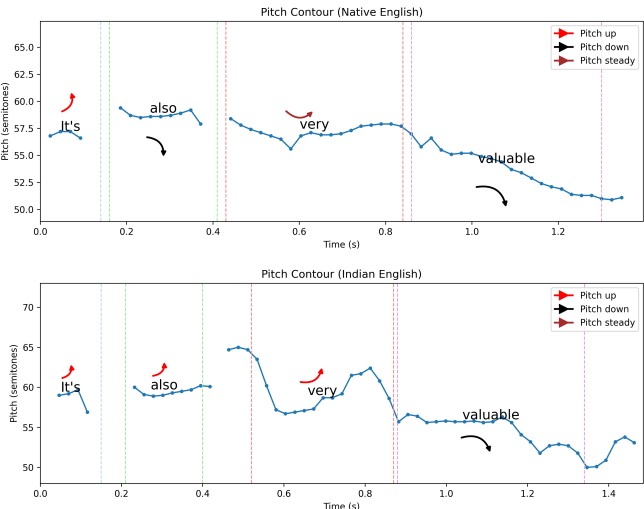

Figure 2: Pitch contour of general American-English accent compared with Indian-English when speaking the sentence "It's also very valuable."

