# OpenReview forum: "Correct and speak: accent reduction with minimum supervision"
_ICLR.cc/2024/Conference — Submitted to ICLR 2024_

### Official Review · Reviewer_Q9U6 · 2023-10-28

**Soundness:** 3 good
**Presentation:** 3 good
**Contribution:** 2 fair
**Rating:** 5
**Confidence:** 3

**Summary:**

This paper proposes a seq-to-seq accent conversion model between Indian and native English. One key idea of the paper is that there should be a semantic token corrector before generating the target audio as there will be some pronunciations in the non-native source audio. The system performs semantic token correction on the HuBERT units and utilizes TFCodec as input to an auto-regressive decoder to generate the target audio.
Experiments suggest that with a smaller model and faster decoding, the proposed model can achieve higher quality outputs as compared to VALL-E-AC approach. The quality is measured in terms of speech naturalness, speaker similarity, and accentedness.

**Strengths:**

+ Originality:
1. The idea of Hubert unit correction to match the non-native accent units to the native ones is a key idea and proves to be important to the model performance
2. Another practical idea about large-scale pre-training and then fine-tuning on a small amount of data could potentially help accelerate developing accent conversion systems for other non-native English accents.

+ Quality:
The paper brings some existing techniques together to solve the accent conversion, it contains a few key ideas and fair evaluation except some concerns mentioned in the Weaknesses section.

+ Clarity: clear enough.

+ Significance: The proposed system can be practical for accent conversion for other non-native English accents.

**Weaknesses:**

My main concern is the reliability of the results and the evaluation protocol.

- Evaluation results show good performance. However, evaluation setups are somewhat weak.
1. For example, subjective listening tests are at a relatively small scale.
2. Evaluating Indian to native English conversion might have better with fluent English listeners who are Indian rather than Chinese to better capture some fine details.
3. It could have been better if the effectiveness of the approach had been shown on a few more non-native English accents.

- Paper layout can be improved. For example, there is a large empty space on Page 8.

- Minor typos need correction.

**Questions:**

1. Did the authors consider other ways of measuring the effect of the influencing factors in Table 1? For example, the edit distance between the Hubert sequences of two files.

2. The subjective analysis in Table 2 has been conducted on a relatively small set of utterances and a small number of evaluators. Could you elaborate more on the reliability of the results in Table 2?

3. References Jiang 2023a and Jiang 2023b are the same.

4. VALL-E has better speaker similarity in Table 3 but not in Table 7. In Table 7 caption, please mention about the zero-shot condition (as opposed to the SPK comparison in Table 3)

5. Minor typing issues:

- Section 4.3.2, first sentence: hubert -> Hubert
- Section 5.1. Baseline models:  Hubert tokens for encoding Employing Hubert tokens for encoding -> please remove repetition
- Appendix is not well-formatted and has some typos (e.g. Accemt -> Accent)

**Details Of Ethics Concerns:**

There is a link to a webpage in the paper, but its content seems to be anonymous as far as I can see.

---

> ### Author Response · Authors · 2023-11-18
> **Author response to Review #4**
>
> We thank the reviewer for the useful comments.
> # Weaknesses:
> **W1**：For example, subjective listening tests are at a relatively small scale.
> **A1**： In the raters' scale, we have increased the number of listeners to 20 to evaluate our model in Sections 3.2 and 5.2. In the test cases, we have also increased the test pairs to 100 from the original 10 in Sec 3.2.  In Sec 5.2, based on previous methods, we believe that 60 cases from 6 Indian speakers are sufficient for our zero-shot test. Subjective tests are costly. For example, in the accent AB evaluation in only the VCTK dataset, a listener needs to listen to 200 cases. That is $C^2_5 \times 20$.
> **W2**：Evaluating Indian to native English conversion might have better with fluent English listeners who are Indian rather than Chinese to better capture some fine details.
> **A2**： Yes, you are right. But that is indeed costly for us, the students. In our test, the raters are college students majoring in foreign languages, who have also been informed about the characteristics of the Indian accent. For example, "Indian-accented speakers may place stress on different syllables, or their intonation may be more flat or elevated, which can make their speech sound more monotone or sing-song to native English listeners." We believe they have the ability to properly conduct accent tests. We also showcase our demo page.
> **W3**：It could have been better if the effectiveness of the approach had been shown on a few more non-native English accents.
> **A3**： Thank you for the advice and i am trying this. We think this framework can be easily extended to other accents with low-resource data.
> **W4**：Paper layout can be improved.Appendix is not well-formatted and has some typos (e.g. Accemt -> Accent)
> **A4**： Sorry about the mistake. We have already made these corrections in our updated paper.
> # Questions
> **Q1**：Did the authors consider other ways of measuring the effect of the influencing factors in Table 1? For example, the edit distance between the Hubert sequences of two files.
> **A1**：The edit distance could also be a good choice. But we think the Longest Common Subsequence Ratio (LCSR) is more intuitive. In Table 1, first, we want to explore the percentage of pure semantic tokens without speaker identity leak, and the percentage of tokens affected by the accent. When the influencing factor is speaker identity, a 75% LCSR indicates that 25% of tokens have speaker identity leakage. A lower probability of speaker leak tokens explains why the generative model works in a zero-shot evaluation. And when the accent influence is added, the LCSR degrades from 0.75 to 0.54, indicating that approximately 21% of the tokens are affected by the accent.
> **Q2**：The subjective analysis in Table 2 has been conducted on a relatively small set of utterances and a small number of evaluators. Could you elaborate more on the reliability of the results in Table 2?
> **A2**： We've expanded to 100 test pairs and 20 raters.
> **Q3**：References Jiang 2023a and Jiang 2023b are the same.
> **A3**： Sorry about this. We have corrected this in our updated paper.
>  **Q4**：VALL-E has better speaker similarity in Table 3 but not in Table 7. In Table 7 caption, please mention about the zero-shot condition (as opposed to the SPK comparison in Table 3)
> **A4**： Our framework has better speaker similarity. In Table 3, the primary reason for lower Speaker Perplexity (SPK) but comparable values to VALLE-AC in the proposed model is that the speech generated by VALLE-AC (without conversion module) retains more accent information. This suggests that more similar speech acoustic information results in more similar speech representations. This is not fair. Therefore, we present Table 7 to compare our proposed generative-only model with VALLE-AC. To avoid misunderstanding, we directly compare our proposed generative-only module with the baselines in Table 3. The results are shown in Table 3.
>  **Q5**：Minor typing issues
> **A5**： Sorry for such typing issues.  We have corrected these.

---

> > ### Comment · Reviewer_Q9U6 · 2023-11-21
> > **Read the Authors' Response**
> >
> > I would like to thank the authors for answering my questions. Also, thanks for increasing the sample size for the subjective tests. However, I would like to keep my previous score.

---

> > > ### Author Response · Authors · 2023-11-22
> > > **Author response to Review #4**
> > >
> > > We thank reviewer #4 for the comments.
> > > As you suggested in **Weakness 3**, we also add more accent cases like Chinese-accent and Korea-accent as the source accent in our demo page which also shows good performance. Please refer to https://convert-and-speak.github.io/demo/.
> > > We have also updated our paper with more clear organization and statements.
> > > We'd appreciate if you can have a second look on these additional updates.

---

> > > > ### Comment · Reviewer_Q9U6 · 2023-11-22
> > > > **Thanks for the response but no score update**
> > > >
> > > > Thanks for additional clarification. However, in a practical application of the proposed system, it would probably be desirable to have high speaker similarity before and after the conversion. Since the results still suggest a slight loss of speaker similarity with the proposed approach, I would like to keep my score as is.

---

> > > > > ### Author Response · Authors · 2023-11-23
> > > > > **Author response to Review #4**
> > > > >
> > > > > We are grateful for the reviewer to have a second look on our new demos on other source accents.
> > > > > With regard to speaker similarity, we dig into the problem of the new demos and find out the slight loss of speaker similarity is caused by the acoustic information in the source accent speech, e.g. reverberation. Since TF-Codec used in this paper is trained on clean speech without reverberation, the tokens generated are dereverbed and may also affect the voice identity. We have added the demos with the dereverbed source accent speech as the prompt for verification. Please refer to ***Prompt with dereverbed source accent speech*** on our demo page(https://convert-and-speak.github.io/demo/). They sound much similar as source speaker which aligns with our testing results on Indian-English to general American-English data set proposed in the paper. Please have a second listen on these cases.

---

### Official Review · Reviewer_o8oG · 2023-10-30

**Soundness:** 3 good
**Presentation:** 3 good
**Contribution:** 3 good
**Rating:** 5
**Confidence:** 4

**Summary:**

This manuscript proposes a one-to-one accent conversion approach, consisting of a seq2seq accent correction model and a waveform generation model. The accent correction model converts hidden units extracted from HuBERTs from non-native speakers to the ones from native speakers. The waveform generation model is implemented based on Codec, and it synthesizes waveform conditioned on converted hidden units and acoustic prompt (which encodes the target speaker identity).

**Strengths:**

* Paper fits ICLR scope well.
* The ideas of using hidden units from HuBERTs as semantic encoding and using Codec based waveform generation models are new. However, the overall framework is mostly the same as prior studies. In summary, the novelty is moderate.
* The proposed method is solid.
* Presentations and references are with good quality.

**Weaknesses:**

* Evaluation and analysis in Section 3 and 5 have limited samples and raters. Issues in implementation and evaluation protocols. (see details in question section)

**Questions:**

Section 3: Please consider comparing hidden units from HuBERT to PPG or other features used in literature. Otherwise the use of hidden units sounds adhoc.

Section 3.1: Please provide more details on the HuBERT setup. How is the model trained? How many clusters are used for hidden units identifications? These have a significant impact on the analysis results. In addition, 10 pairs don’t seem to be enough to have the conclusions on the impact of accent and speaker identity from hidden units. Please verify this on a larger dataset.

Section 3.2: Similarly, the results are less convincing with limited human ratings. Please consider increasing the number of pairs and the number of raters.


Sample page: Please list an utterance that is used for acoustic prompting as well here.

Section 5.2: In VCTK experiments, please use more source speakers instead of just one.

Section 5.2: Accent: “During the AB test on accent score, the ground truth (GT) speech samples were selected from the native speaker ”bdl” in the L2-ARCTIC dataset.” Is it still the case when the non-native speaker is a female? If so, please consider using a female voice as the reference.

Section 5.4: Just curious but not required - Does the proposed correction work under one-shot or zero-shot setup? 15-mins of parallel data is already a lot in production scenarios.

Section 5.4.1: What about accent and naturalness of the zero-shot setup?

---

> ### Author Response · Authors · 2023-11-18
> **Author response to Review #3**
>
> Dear reviewer, thanks a lot for your review.
> **Q1**：Section 3: Please consider comparing hidden units from HuBERT to PPG or other features used in literature. Otherwise the use of hidden units sounds adhoc.
> **A1**：In our framework, we require a speech tokenizer that can extract highly semantic, highly compressed tokens. It has been demonstrated that HuBert model satisfies these requirements [1][2][3]. On the other hand, PPG is a frame-level feature that involves more speaker information leakage, making it more suitable for mapping tasks. The baseline model in AC-PPG illustrates that mapping from L2 PPGs(non-native feature) to L1 PPGs(native feature) may not be as effective.
> **Q2**：Section 3.1: Please provide more details on the HuBERT setup. How is the model trained? How many clusters are used for hidden units identifications? These have a significant impact on the analysis results. In addition, 10 pairs don’t seem to be enough to have the conclusions on the impact of accent and speaker identity from hidden units. Please verify this on a larger dataset.
> **A2**：Thank you for pointing this. We provide additional details in Section 3.1: "we utilize the HuBERT-Base (Hsu et al., 2021) model and a k-means algorithm with 500 clusters to extract HuBERT tokens". The pretrained model can be accessed at https://huggingface.co/facebook/hubert-base-ls960. Additionally, in Section 3.1, we've expanded to 1000 test pairs from 10 pairs. We arrived at similar conclusions, and our experimental results and analyses have been publicly commented and updated in the paper.
> **Q3**：Section 3.2: Similarly, the results are less convincing with limited human ratings. Please consider increasing the number of pairs and the number of raters.
> **A3**：In Sec 3.2, we've expanded to 100 test pairs and 20 raters.
>  **Q4**：Sample page: Please list an utterance that is used for acoustic prompting as well here.
> **A4**：The utterance used for acoustic prompting is also derived from the source non-native speech. And we also add these in sample page. In our framework, we randomly select 3-second segments from the source speech to extract acoustic prompts, aiming to maintain the same speaker identity post-accent conversion. If the original speech is less than 3 seconds, we take the whole sequence of the original speech.
>  **Q5**：Section 5.2: In VCTK experiments, please use more source speakers instead of just one.
> **A5**：We follow our baseline model AC-PPG‘s evaluation setting, which utilizes 20 test cases from a speaker in the VCTK dataset which is not parallel dataset. To explore why our model performs effectively through the LCSR metric, we require the parallel datasets like Arctic and L2-Arctic datasets.
>  **Q6**：Section 5.2: Accent: “During the AB test on accent score, the ground truth (GT) speech samples were selected from the native speaker ”bdl” in the L2-ARCTIC dataset.” Is it still the case when the non-native speaker is a female? If so, please consider using a female voice as the reference.
> **A6**：As A4, the utterance used for acoustic prompting come from the source non-native speech to maintain the same speaker identity. The native speaker, 'bdl', serves as the ground truth speaker in our accent metric.
>  **Q7**：Section 5.4: Just curious but not required - Does the proposed correction work under one-shot or zero-shot setup? 15-mins of parallel data is already a lot in production scenarios.
> **A7**：Our framework operates under zero-shot setup. The six test speakers are unseen by our generative module, and the conversion module has only been exposed to an Indian speaker. We do not require audios with the same content but from the same speakers with different accents. The speaker identities can be different. Furthermore, we are able to acquire 4 hours of such parallel data from the Arctic and L2-Arctic datasets.
>  **Q8**：What about accent and naturalness of the zero-shot setup?
> **A8**：As A7, our evaluation operates under zero-shot setup.
> [1] Hsu, Wei-Ning, et al. "Hubert: Self-supervised speech representation learning by masked prediction of hidden units." IEEE/ACM Transactions on Audio, Speech, and Language Processing 29 (2021): 3451-3460.
> [2] Wang, Zhichao, et al. "LM-VC: Zero-shot Voice Conversion via Speech Generation based on Language Models." arXiv preprint arXiv:2306.10521 (2023).
> [3]Huang, Rongjie, et al. "Make-A-Voice: Unified Voice Synthesis With Discrete Representation." arXiv preprint arXiv:2305.19269 (2023).

---

> > ### Comment · Reviewer_o8oG · 2023-11-21
> > **Thanks for the clarifications!**
> >
> > I would like to thank the authors for detailed explanations to my questions and the improvements made to the paper, and I increase my soundness scores accordingly. Given the moderate novelty and the issues pointed out by other reviewers, I would keep the overall rating as it is.

---

> > > ### Author Response · Authors · 2023-11-22
> > > **Author response to Review #3**
> > >
> > > Dear reviewer, thanks for your comment.
> > > We also add more accent cases like Chinese-accent and Korea-accent as the source accent in our demo page which also shows good performance. We highly recommend you to have a listen on https://convert-and-speak.github.io/demo/.
> > > For the issues posted by other reviewers. We have updated them in our paper quickly. The new paper is updated with more clear organization and statements.
> > > We'd appreciate if you can have a second look on these additional updates.

---

### Official Review · Reviewer_SdoC · 2023-11-02

**Soundness:** 3 good
**Presentation:** 3 good
**Contribution:** 3 good
**Rating:** 8
**Confidence:** 4

**Summary:**

* This paper proposes a new framework for accent conversion (AC) with minimum supervision. The framework consists of a correction module and a speaking module. The correction module corrects the source accented semantic tokens to the target native ones, while the speaking module generates speech with native prosody and the same speaker identity. The correction module is pre-trained with a pretext task in a self-supervised manner using large amounts of native speech and finetuned with only 15 minutes of parallel data. Experimental results show that the proposed framework achieved the state-of-the-art performance in terms of accentedness, speech quality and speaker maintenance.

* The manuscript includes a non-anonymous github link for the samples (https://jiazj-jiazj.github.io/Speak-and-correct/).

**Strengths:**

The strengths of the paper are as follows:

- originality: The paper proposes a framework based on generative models for accent conversion with minimum supervision. Accent Conversion is an existed task, and the author try to address two challenges including: 1) training with less parallel data, 2) removal of accent effects on prosody and pronunciation patterns. Although pre-training with large unlabeled data and finetuning with a few parallel data is not a creative idea (e.g., Spear-TTS), the paper does effectively solve the two problems, and the authors have comprehensively analyzed and verified the relationship between accent, semantic tokens and acoustic tokens. Besides, introducing TF-Codec instead of Encodec/SoundStream shows the improvement in reducing the complexity and latency, which is helpful for speech generation.

- quality: The quality of the paper is high, with clear problem definition, adequate literature review, comprehensive analysis and well-organized presentation.

- clarity: The paper is generally well-written and easy to follow.

- significance: The significance of the paper is that it provides a novel method for accent conversion that can be used with a small amount of parallel data. The similar idea of decomposing accent conversion into semantic token generation and acoustic token generation has been attractive recently, which can be inspiring for other works in the field of speech generation.

Overall, the paper is a relatively valuable contribution to the field of accent conversion and speech generation.

**Weaknesses:**

The weakness of the paper are as follows:

* From the perspective of method, introducing TF-Codec as a contribution seems independent and unrelated to the theme of paper (accent conversion or accent reduction). If the speaking module is replaced with a multi-stage generative model such as Encodec, will it affect the accentedness and speaker similarity of accent conversion?

* Although generally well-written, there exists some unclear and confusing statements in the paper. Some technical details and discussions are missing and need to be included. Please checkout the questions below.

**Questions:**

* Does it remove all the dupilicated semantic tokens in HuBERT in both correction module and speaking module as stated in Section 3.1?

* The formula definition in “Pretraining” in Section 4.2. What is the purpose of defining $C^{t-1}$? It seems to be not used. Does $X^{t-1}$ already include $x_t$ or not? This part may be roughly understood, but it is not clear enough.

* In Section 4.2, what is the token mask ratio and strategy in pretraining stage? These details are important to reproduce the paper. Please give more explanations.

* According to Section 4.3, the input of speaking module is the concatenation of the prompt accented semantic tokens, the target native semantic tokens, and the prompt accented acoustic tokens during inference, right? If so, the combination of accented semantic tokens and native semantic tokens has not been seen when training the speaking module. Does this mismatch affect the performance of accent conversion?

* In section 5, what is the difference between “proposed” and “proposed-VC”? Are they the same one? Is “the traditional AC model (Liu et al., 2020)” equal to “AC-PPG”? It should be better to present the name and settings of each model more clearly.

---

> ### Author Response · Authors · 2023-11-18
> **Author response to Review #2**
>
> We thank the reviewer very much for recognizing our work and suggestions. This is the final version of our comment.
> # Weaknesses:
> **W1**：From the perspective of method, introducing TF-Codec as a contribution seems independent and unrelated to the theme of paper (accent conversion or accent reduction). If the speaking module is replaced with a multi-stage generative model such as Encodec, will it affect the accentedness and speaker similarity of accent conversion?
> **A1**： TF-Codec is the core design of our generative model which enables a single-stage causal speech generation. Compared with multi-stage speech generation based on EnCodec, it achieves lower complexity and has higher potential for low-latency applications. Besides complexity, TF-Codec generates speech with higher speaker similarity (0.502 vs 0.429 in terms of SPK）and naturalness (4.08 vs 3.84 in terms of MOS) in ***Table 3***.
> # Questions
> **Q1**：Does it remove all the duplicated semantic tokens in HuBERT in both correction module and speaking module as stated in Section 3.1?
> **A1**：In the training and inference phase, we don't remove duplicated HuBERT semantic tokens. Somehow, it helps to retain the original speech duration.  In ***Section 3.1***,  we remove duplicated semantic tokens to eliminate the effect of the duration of each word in the accent analysis on HuBERT tokens.
> **Q2**：The formula definition in “Pretraining” in Section 4.2. What is the purpose of defining $C^{t-1}$? It seems to be not used. Does
> $X^{t-1}$ already include $x_{t}$or not? This part may be roughly understood, but it is not clear enough.
> **A2**：Sorry for the mistake. We correct the formula in our paper.  Please refer to ***Formula (1)*** in ***Section 4.2***.
> **Q3**：In Section 4.2, what is the token mask ratio and strategy in pretraining stage? These details are important to reproduce the paper. Please give more explanations.
> **A3**：We have added an explanation in our paper in ***Section 4.2 Pretraining*** as ***"We have experimented with corruptions like token masking, token deletion, and token infilling and we find the masking scheme works the best. Specifically, following the masking scheme in BERT, 15\% tokens have been randomly selected first. Then 80\% of them are replaced with [MASK] tokens, 10\% of them are filled with random tokens and the remaining 10\% are left unchanged.".***
>  **Q4**：According to Section 4.3, the input of speaking module is the concatenation of the prompt accented semantic tokens, the target native semantic tokens, and the prompt accented acoustic tokens during inference, right? If so, the combination of accented semantic tokens and native semantic tokens has not been seen when training the speaking module. Does this mismatch affect the performance of accent conversion?
> **A4**：That's a good question. The setting is similar to ***Section 3.2***. In ***Section 3.2***, the generative model is trained to generate speech conditioned on native semantic tokens. During inference, the semantic token part is the concatenation of the native content semantic tokens and the prompt semantic tokens. The experiment compares the native prompt semantic tokens with the accent semantic tokens and the result shows similar which indicates such mismatch doesn`t affect the performance. This can also be verified in the performance of our current model on AC. Please refer to our demo: https://convert-and-speak.github.io/demo/.
> **Q5**：In section 5, what is the difference between “proposed” and “proposed-VC”? Are they the same one? Is “the traditional AC model (Liu et al., 2020)” equal to “AC-PPG”? It should be better to present the name and settings of each model more clearly.
> **A5**：***Proposed-VC*** in Section 5 in previous paper is ***Proposed*** without conversion module. Sorry for the unclear expression. We correct these in ***Table 3*** in the updated paper, in which ***Proposed-VC*** refers to ***Generative model(TF-Codec)*** and ***Liu. et al*** refers to existing best machine-learning based AC method available in the public, previously named  ***AC-PPG***.

---

> > ### Comment · Reviewer_SdoC · 2023-11-23
> > **Thanks for Authors' responses**
> >
> > Thank the authors for detailed explanations and revisions to the paper in response to my questions.  I have also read the other reviewers' comments and authors' responses.  I would keep my original rating.

---

### Official Review · Reviewer_KUHy · 2023-11-05

**Soundness:** 3 good
**Presentation:** 2 fair
**Contribution:** 3 good
**Rating:** 6
**Confidence:** 5

**Summary:**

This paper presents an accent conversion model, composed of two major components: 1) an correction module for converting the accent in a discrete latent domain; 2) a generation module for generation speech features. The generated speech features are discrete tokens that can be converted into audio waveforms with a neural vocoder.

The experiment was conducted by training the model on LibriTTS + ARCTIC + L2 ARCTIC, and evaluating on 5 speakers from L2 ARCTIC and another speaker from VCTK.

**Strengths:**

The presented method is sound.

**Weaknesses:**

- The presentation and writing needs improvement.
- There are concerns on technical correctness, see "Questions".
- There are concerns on discrimination, see "Ethics Review". My initial rating is primarily based on such concerns, and was updated after the authors addressed them.

**Questions:**

1. Abstract: mentioned three terms: "accent conversion", "accent removal", and "accent reduction". It would be helpful to distinguish or consolidate.
2. Abstract: "TF-Codec" used without explanation nor reference.
3. Sec 2: "there has not been a parallel corpus that contains pairs of audios having the same contents yet coming from the same speakers in different accents" -- better to restrict such claims as "public corpus", as you don't know proprietary ones.
4. Sec 3.1: `LCSR = LCS / utterance_length` -- clarification is needed because the two utterance can have different lengths.
5. Sec 3.1: Table 1 is misleading because "accent without phoneme corruption" and "accent with phoneme corruption" implies different speaker. So the trends in the table is completely expected. It cannot draw conclusion as the paper stated, that " the speaker identity causes little impact on the Hubert tokens and the content is not disturbed", "accent feature ... brings larger influence on the Hubert tokens".
6. Sec 3.2: which model is used for synthesis? this section looks like experimental results, rather than analysis.
7. Sec 5.1: any explanation on why the same speaker "ASI" is used for both training and evaluating?
8. Sec 5.1: "In the AB test for accenteness, participants first listened to native and non-native reference audio. Subsequently, they heard paired speech samples (A and B) with identical content and were asked to choose the one that sounded more native-like." -- will rater be able to infer from the voice identity instead of accent?
9. Sec 5.4: This seems completely unreadable to me. I don't understand what the sentences mean, and have no idea about what "3.5, 3, 2.7" numbers refer to as they are not presented in the corresponding Table.
10. Sec 5.4.1: Does the 2% LCSR in Table 6 consistent with 54% LCSR in Table 1?

**Details Of Ethics Concerns:**

I'm concerned on the way that this paper describes about foreign accents. It describe foreign accents as "corrupted" or "distorted" pronunciation, and what the proposed model does is "correcting". This is discriminative and should be avoided. There are neutral words can be used, such as "translate", or "convert", instead of "correct".

---

> ### Author Response · Authors · 2023-11-18
> **Author response to Review #1 （1/2）**
>
> We thank the reviewer for the useful advice.
> **About Ethics Concerns**
> **A**：We are very grateful to the reviewer for pointing out the ethics issues in our work. We have revised the wrong description such as "distorted" pronunciation updated to "different" pronunciation in foreign accents. We use the term 'convert' instead of 'correct' in our paper, including title, abstract and main parts.
> **Q1**：Abstract: mentioned three terms: "accent conversion", "accent removal", and "accent reduction". It would be helpful to distinguish or consolidate.
> **A1**：  Thank you for advising this. We refer to these as 'accent conversion' in our updated paper.
> **Q2**：Abstract: "TF-Codec" used without explanation nor reference.
> **A2**： TF-Codec is a speech neural codec which follows paper[1]. We update this in abstract "*TF-Codec is a pretrained speech neural codec with group quantization which can be used as a single-stage autoregressive generation.*"
> **Q3**：Sec 2: "there has not been a parallel corpus that contains pairs of audios having the same contents yet coming from the same speakers in different accents" -- better to restrict such claims as "public corpus", as you don't know proprietary ones.
> **A3**：We update this "*For accent conversion task, there has not been a public parallel corpus that contains pairs of audios having the same contents yet coming from the same speakers in different accents.*"
> **Q4**：Sec 3.1: LCSR = LCS / utterance_length -- clarification is needed because the two utterance can have different lengths.
> **A4**：We update this "*The length of the parallel data in each pair is similar and the utterance length ranges from 3 seconds to 5 seconds. The smaller utterance length is used to calculate the LCSR of each pair.*"
> **Q5**：Sec 3.1: Table 1 is misleading because "accent without phoneme corruption" and "accent with phoneme corruption" implies different speaker. So the trends in the table is completely expected. It cannot draw conclusion as the paper stated, that " the speaker identity causes little impact on the Hubert tokens and the content is not disturbed", "accent feature ... brings larger influence on the Hubert tokens".
> **A5**： We update the description in my paper “As the results shown, with the accent source, HuBert tokens have changed a lot, degrade from 0.75 to 0.57 in terms of LCSR. For those cases with specific phoneme changes, more tokens have changed
> from their native references(LCSR: 0.54).”
> [1] Jiang, Xue, et al. "Latent-Domain Predictive Neural Speech Coding." IEEE/ACM Transactions on Audio, Speech, and Language Processing (2023).

---

> ### Author Response · Authors · 2023-11-18
> **Author response to Review #1 （2/2）**
>
> **Q6**：Sec 3.2: which model is used for synthesis? this section looks like experimental results, rather than analysis.
> **A6**：The generative model in our proposed framework is used for this analysis experiment. In this section, we evaluate if the accent feature will be extended through the in-context learning. Based on this analysis, we find the accent style can hardly be transferred through the in-context learning in the generative model, which guides us to use the original accent source as the prompt to maintain the speaker identity.
> **Q7**：Sec 5.1: any explanation on why the same speaker "ASI" is used for both training and evaluating?
> **A7**: We utilize the speaker "ASI" for training and use 6 Indian speakers from VCTK, ARCTIC, and L2-ARCTIC for evaluation. We have now corrected this in Sec 5.1“*In the process of fine-tuning the correction model, we utilize data from speaker "bdl" as the native English speaker source ...... and data from speaker "ASI" ....... In the evaluation, we employ data from speaker p248 originating from the VCTK dataset, as well as data from five speakers, specifically ASI, KSP, RRBI, and SVBI, derived from the L2-ARCTIC dataset to assess the performance of models.*”
> **Q8**：Sec 5.1: "In the AB test for accenteness, participants first listened to native and non-native reference audio. Subsequently, they heard paired speech samples (A and B) with identical content and were asked to choose the one that sounded more native-like." -- will rater be able to infer from the voice identity instead of accent?
> **A8**:  The native and non-native reference audio initially listened to have different speaker identities compared to the test pairs. Or if you're asking whether the speaker's identity is related to the accent in the test cases? Before the test, the raters have been taught to discern differences in Indian accents. And we direct the speakers to focus solely on accent differences.  These 5 speakers are all indian- accent speakers in l1-l2 arctic dataset.
> **Q9**：Sec 5.4: This seems completely unreadable to me. I don't understand what the sentences mean, and have no idea about what "3.5, 3, 2.7" numbers refer to as they are not presented in the corresponding Table.
> **A9**:  Sorry for this mistake. We have already deleted this.
> **Q10**：Sec 5.4.1: Does the 2% LCSR in Table 6 consistent with 54% LCSR in Table 1?
> **A10**: Apologies for the misunderstanding due to the lack of explanation. In Table 6,  "without correction model," means that in  the finetuning stage, we directly use the L2-semantic token and L1-acoustic tokens with different speaker identities but with the same phoneme content to fine-tune the generative model instead of using a converted model to convert non-native semantic tokens to native semantic tokens. 2% LCSR means it does not work.

---

> > ### Comment · Reviewer_KUHy · 2023-11-22
> >
> > Thanks for the responses from the authors. A lot of my questions are answered clearly.
> >
> > I have a few follow up questions:
> >
> > A5: My concern remains, because the table still implies that the first row is the impact of speaker identity, and the last two rows are the impact of the accent. It should be made clear in the table that the last two rows are comparison across different speakers as well.
> >
> > A6: From the structure of the writting, Sec 3.2 is supposed to be a pre-analysis to lay out the ground and the motivation of the approach, but it's actually analysis on the proposed method. It fits better in the "Experiments" section.
> >
> > A7: So speaker "ASI" is used in both training and evaluation indeed. What's the impact in the evaluation results and the explanations?
> >
> > A8: According to the paper, "the ground truth (GT) speech samples were selected from the native speaker ”bdl”". In the meantime, to model outputs to be evaluated sounds clearly synthesized (because of artifacts or other factors results into lower naturalness). So it would be quite clear to the raters which audio samples is supposed to be the "native" version (true recording of speaker bdl) and which audio sample is supposed to be "more accented" (synthesized, may or may not be in a different voice). Such clue can crew the side-by-side evaluation result drastically, despite that you "direct the speakers to focus solely on accent differences".
> >
> > A10: What's the reason of that ("it does not work")? A doing-nothing model should get 54% LCSR here based on Table 1, right?

---

> > > ### Author Response · Authors · 2023-11-22
> > > **Author response to Review #1**
> > >
> > > Dear reviewer, thank you very much for taking the time to read and respond to our previous reply.
> > > **Question-A5**: My concern remains, because the table still implies that the first row is the impact of speaker identity, and the last two rows are the impact of the accent. It should be made clear in the table that the last two rows are comparison across different speakers as well.\
> > > **Answer-A5**: Yes, you are right. The last two rows are comparison across different speakers. The speaker identity effect cannot be eliminated in the accent analysis since we can not find the strict parallel data with the same speaker speaking in two accents. So the first row which is an LCSR test on speaker identity is working as a reference. We have made this clear in our updated paper ***"Since the speakers of each pair are different, the effects of LCSR on the speaker identity is also calculated as a reference"*** and ***Table.1*** in Section 3.1.
> > >
> > > **Question-A6**: From the structure of the writting, Sec 3.2 is supposed to be a pre-analysis to lay out the ground and the motivation of the approach, but it's actually analysis on the proposed method. It fits better in the "Experiments" section.\
> > > **Answer-A6**：Yes, we need to evaluate if the accent feature will be extended through the in-context learning with generative model before we feed the accent source speech into the generative model directly to get the speaker identity. This is the foundation of our design to maintain the source speaker identity and avoid bringing the accent back in the converted speech by using like this. The generative model is not the new work we proposed in this paper. Actually any existing generative models, e.g. EnCodec-based generative model can be used in this analysis. For simplicity, we use the proposed TF-Codec based generative model at hand to do such analysis. Similar results should be reached. Thanks for your suggestion. It should be better to use an existing model to do the pre-analysis.
> > >
> > > **Question-A7**: So speaker "ASI" is used in both training and evaluation indeed. What's the impact in the evaluation results and the explanations?\
> > > **Answer-A7**: To clarify, for speaker "ASI", we use 1000 utterances in the training and 100 utterances in the testing. Besides this speaker, we also use other speakers which is not seen in training to verify the zero-shot ability. Sorry for the unclear statements in previous paper, we have updated ***Section 5.1 Dataset*** in the paper.
> > >
> > >
> > > **Question-A8**: According to the paper, "the ground truth (GT) speech samples were selected from the native speaker ”bdl”". In the meantime, to model outputs to be evaluated sounds clearly synthesized (because of artifacts or other factors results into lower naturalness). So it would be quite clear to the raters which audio samples is supposed to be the "native" version (true recording of speaker bdl) and which audio sample is supposed to be "more accented" (synthesized, may or may not be in a different voice). Such clue can crew the side-by-side evaluation result drastically, despite that you "direct the speakers to focus solely on accent differences".\
> > > **Answer-A8**: We think the authenticity bias could be avoided in our experiment design. First, we use A/B testing instead of Comparison Category Rating (CCR) subjective testing in which any pairs will be taken into the comparison besides the \<converted speech, GT\>. The source accent speech which is also the true recording is also included in the testing. Secondly, it is not so easy to make the judgement just depending on the naturalness since the converted speech from these models are mostly with high naturalness according to the MOS-Naturalness subjective testing.
> > >
> > > **Question-A10**: What's the reason of that ("it does not work")? A doing-nothing model should get 54% LCSR here based on Table 1, right?
> > > **Answer-A10**: Sorry for the misclarification on 2% LCSR in ***Table.7 (in the updated paper)*** and 54% LCSR in ***Table.1***. To clarity, 2% LCSR is the model finetuned on the pretrained generative model with a small amount of parallel data. Specifically, when using the generative model pretrained on large native speech corpus, 54% LCSR can be achieved, indicating the model is working for normal speech generation but bad accent conversion, e.g. accurate content but accent prosody. However, when we use a small amount of parallel data to finetune this model, we find the model fails to learn the acoustic mapping between accent semantic tokens and native acoustic tokens, which results in totally wrong speech, e.g. 2% LCSR. We add more explanations in ***Section 5.5*** in the updated paper as ***"This solution totally fails as the 2.0\% LCSR in Table.7 shown, indicating hardly any accurate content preserved through this mapping-based learning method with little amount of parallel data."***.

---

> > > > ### Comment · Reviewer_KUHy · 2023-11-22
> > > >
> > > > Thanks for the responses and additional revisions, especially on address the ethics concerns. I have updated my overall rating score from 1 to 6, and contribution from 2 to 3.
> > > >
> > > > Although the evaluation protocol and the experiment set up in the paper has a room for improvement (Q7 and Q8), I believe the current form is informative and helpful to the community. I'd appreciate if the authors can continue revise the text to clarify the limitations of the evaluation and the results.

---

> > > > > ### Author Response · Authors · 2023-11-23
> > > > > **Author response to Review #1**
> > > > >
> > > > > We are grateful for the reviewer to have a quick response and informative discussion with us. Definitely, we will further improve the evaluation framework of accent conversion task by enlarging the testing scale and a more normative subjective evaluation framework in our future research. We also add this commitment in the ***Conclusions*** in the updated paper.

---

### Author Response · Authors · 2023-11-18
**Common Response-Summary of Updated Version**

Thank all reviewers for your suggestions. We take them seriously. This is the final version of our public comment. Please refer to this one.
1. Thanks for pointing out the ethics concerns. We use the term ***'convert'*** instead of ***'correct'*** in all related descriptions. Technically, the proposed framework can be extended to many-to-many accent conversion tasks. So it is more accurate to use ***'convert'***.  In this paper, we are just starting with Indian-accent. And we have added more accent cases like Chinese-accent and Korea-accent as the source accent in our demo page which also shows good performance. Please refer to https://convert-and-speak.github.io/demo/.
2. For experiments, basically we enlarge the testing scales. Specifically, in ***Section 3.1***, we increase to 1000 testing pairs from 10 testing pairs. In ***Section 3.2***, we increase to 100 testing pairs from 10 testing pairs and 20 raters from 8 raters.  The experimental results have been updated in the paper and similar conclusions can be reached.
3. More accurate names are given for the baseline models in ***Table 3***  :
    - ***Liu. et al***, the existing best machine-learning based AC method available in the public, previously named  ***AC-PPG***
    - ***Generative model(EnCodec)***, a generative model based on EnCodec, previously named ***VALLE-AC***
    - ***Generative model(TF-Codec)***, a generative model based on TF-Codec,  the proposed framework without conversion module, previously named ***Proposed-VC***
4. Paper re-organization and typing issues update. Very sorry about this. Please refer to our updated paper.

---

### Comment · Area_Chair_AjJJ · 2023-11-21
**Reminder to reviewers to participate in the author/reviewer discussion**

Dear reviewers, this is a reminder that the author/reviewer discussion period ends November 22.

This discussion is indeed supposed to be a dialog, so please respond to
the comments from the authors as well as the updated manuscript.

AC

---

### Meta-Review · Area_Chair_AjJJ · 2023-12-05

**Metareview:**

## Scientific Claims and Findings
This paper proposes a model for accent conversion that relies on a sequence-to-sequence model to convert semantic tokens extracted from speech in the source accent using a HuBERT model to semantic tokens appropriate for the same linguistic content rendered in the target accent and an autoregressive generative model that drives a neural codec to synthesize speech in the target accent given a (converted) sequence of semantic tokens and a speech prompt from the source speaker to preserve speaker identity. **A central claim of the paper (Section 3.2) is that this speech prompting preserves speaker identity in the generated speech, but that it does not preserve accent.** The neural codec used in this work is a recently proposed one that generates multiple tokens in a single step, allowing for more efficient inference. To minimize the need for parallel data, the semantic token conversion model is initially trained on large quantities of data in the target accent and then fine-tuned with smaller amounts of parallel data. The speech generation model is likewise trained on large quantities of data in the target accent. The paper analyzes data from the L1-L2 ARCTIC dataset to show that accent affects HuBERT tokenizations. Comparisons using an objective measure of speaker similarity and subjective measures of naturalness and accent similarity are made to show that the proposed model outperforms three baselines.

## Strengths
- The convert-and-generate architecture is a reasonable one to explore for this task.
- The analysis showing that HuBERT tokenization is affected by accent is valuable.

## Weaknesses
- Performing judgements of accent using a panel of listeners who are not native speakers of the target accent is a serious methodological flaw. Accent is manifested in the speech signal via a wide variety of different factors, including phonetic substitutions, allophonic variations, differences in tempo and pause patterns, differences in pitch tracks, and differences in timbre. It is extremely difficult for non-native listeners to properly check all these sources of variation. Because the paper hinges on the very strong claim made in Section 3.2 that accent does not transfer from a prompt into synthesized speech, and this claim is made based on listening by a panel of non-native speakers of American English, I do not consider the results of the paper to be reliable.
- I went to the samples and listened to a number of them, and observed that there are still strong indications of the source accent in the test cases. For example, in the "Six spoons of fresh snow peas..." sample, the "th" in "thick" is rendered with an aspirated "t", which is quite typical of Indian-accented English, but not of American English, and the extremely long pause after "for her" and before "brother Bob" is also highly unusual in American-accented English and quite frequent in Indian-accented English.
- The paper does not provide confidence intervals for the MOS results.

**Justification For Why Not Higher Score:**

- The use of non-native speakers to make subjective judgements about accent is a serious methodological flaw.
- The claim that one can transfer speaker identity, but not speaker accent, from a speech prompt in the generative model is an extremely strong one that needs to be carefully substantiated, and the paper simply doesn't do that.

**Justification For Why Not Lower Score:**

N/A

---

### Decision · Program_Chairs · 2024-01-16

Reject